# Leveraging Telehealth for the Management of Breast Cancer: A Systematic Review

**DOI:** 10.3390/healthcare10102015

**Published:** 2022-10-12

**Authors:** Clemens Scott Kruse, Gerardo J. Pacheco, Brea Vargas, Nadya Lozano, Sergio Castro, Manasa Gattu

**Affiliations:** School of Health Administration, Texas State University, 601 University Drive, San Marcos, TX 78666, USA

**Keywords:** mHealth, telemedicine, breast cancer

## Abstract

Background: Breast cancer affects 2.3 million women and kills 685,000 globally, making it the most prevalent cancer. The telemedicine modality has been used to treat the symptoms associated with breast cancer recovery. Objectives: To analyze the effectiveness of telemedicine to help women recover from the treatment-associated effects and promote overall recovery from breast cancer. Methods: Four databases were queried for published literature from the last 10 years. The systematic literature review was conducted in accordance with the Kruse Protocol and reported in accordance with PRISMA 2020. Results: Five interventions were identified in the literature, with the most dominant being eHealth and mHealth. The other interventions were telephone, video teleconference, and a combination of eHealth and mHealth. There were positive effects of these telemedicine interventions in 88% of the studies analyzed. Telemedicine is shown to positively affect physical and mental health, sleep outcomes, quality of life, and body image. The largest barriers to the adoption of telemedicine interventions are training, cost, workflow, time of providers, and low reimbursement. Conclusion: Telemedicine offers promise to both providers and breast cancer survivors to improve the physical and mental health detriments of both cancer and its associated treatments. It also helps women develop healthy habits to reduce the risk of reoccurrence.

## 1. Introduction

### 1.1. Rationale

Breast cancer is a disease, originating in the breast, in which breast cells grow out of control [1]. The incidence of breast cancer is extensive. In 2020, for example, over 2.3 million women were diagnosed with this condition, and this resulted in 685,000 deaths globally. The 5-year prevalence was estimated at 7.8 million women, which establishes it as the world’s most prevalent cancer [2]. Breast cancer treatment is effective when caught early. Treatment often includes surgical removal, radiation therapy, and medication, but all of these treatments come at a physical and emotional cost to the survivor. Providers have sought new and innovative means to help women through the treatment process and the aftermath of the emotional devastation it brings. Telemedicine offers some interventions.

Telemedicine is defined as healing at a distance through the use of information and communications technologies (ICT) [3]. Telemedicine takes on many forms, but in general, it provides clinical support and overcomes geographical boundaries to improve health outcomes through ICT. Although many distinguish between telehealth and telemedicine, the World Health Organization does not distinguish between them, therefore, telehealth and telemedicine will be used interchangeably in this study. One form of telemedicine is mHealth and eHealth, or mobile-based health and computer-based health, respectively. These take the form of mobile apps, text messages through short message service (SMS), telephonic calls, websites, and computer programs. Many eHealth interventions can now be accessed on mobile devices, therefore the lines between the modalities have become blurred.

Several forms of telehealth have been used for the last several years in the area of oncology, and specifically breast cancer. mHealth apps have shown effectiveness in improving mood, symptom interference, self-efficacy, self-esteem, and emotional functioning [4]. mHealth apps provide education and improve health literacy [5,6]. They improve medication adherence and help women with coping strategies [7,8]. Overall, mHealth apps have shown positive effects on the perception of physical benefits, psychological factors such as motivation, social factors such as group practice, and organizational factors including preplanning physical activity [9]. The paucity of evidence for clinical efficacy begs additional research. This is the justification for this study.

In 2021, a systematic review was published examining mHealth interventions’ ability to improve the quality of life for cancer patients. They identified 25 articles over a period of 10 years. They found the most common issues addressed by mHealth were physical activity, mindfulness, and stress management. Overall, mHealth had a positive effect on patients [10].

In 2022, a scoping review was published that examined mHealth’s ability to increase screening rates among Hispanic communities. Ten articles were selected out of an original result of 597 from a search that spanned ten years. The reviewers reported mHealth was effective at providing education and increasing health literacy [6].

### 1.2. Objectives

The purpose of this review is to analyze the effectiveness of telehealth interventions to manage breast cancer care and recovery.

## 2. Methods

### 2.1. Eligibility Criteria

To be included in the group of articles for analysis, studies had to be published in the English language in the last 10 years in peer-reviewed, academic journals, and used human adult females as subjects. To avoid confounding results, other reviews were excluded. Systematic reviews summarize the findings of previous results (from a set number of years). Including a systematic review from 2022 in the analysis, for instance, would include results from articles also analyzed separately. This would double count instances of findings, which would confound the results.

### 2.2. Information Sources

Four databases were queried: The U.S. Library of Medicine’s PubMed (MEDLINE), the Cumulative Index of Nursing and Allied Health Literature (CINAHL), Web of Science, and Embase’s Science Direct. These databases were searched on 2 August 2022. We also performed a journal-specific search of Healthcare.

### 2.3. Search Strategy

We used the U.S. Library of Medicine’s Medical Subject Headings (MeSH) to create a Boolean search string to combine key terms into an exhaustive search: (mHealth OR telemedicine OR “mobile apps”) AND (“breast cancer” AND “treatment”). The same search string was used in all databases, and as much as possible, we used the same filters in each database. MEDLINE was excluded from all databases except PubMed since PubMed includes the MEDLINE database. This action helped eliminate duplicates.

### 2.4. Selection Process

Search results were filtered and abstracts were screened in accordance with the Kruse Protocol [11] and reported in accordance with PRISMA 2020 [12]. The Kruse Protocol was written to demonstrate the veracity of using the systematic literature review in higher education, but it outlines a proven methodology that has been published over 50 times in high-quality journals [11]. The PRISMA 2020 standard provides a systematic methodology to ensure standardized fields are reported for all systematic reviews and meta-analyses. Abstracts were screened by at least two reviewers.

### 2.5. Data Collection Process

An Excel spreadsheet, standardized in the Kruse Protocol, was utilized as a data extraction tool, collecting additional data at each step of the process. Three consensus meetings were held to identify articles for analysis, perform a narrative or thematic analysis, and perform additional analysis on the results to identify trends [11,13]. Abstracts were screened and studies were analyzed by at least two reviewers throughout the process.

### 2.6. Data Items

We collected the following fields of data: research database source, year of publication, authors, title of study, journal, study participants, experimental intervention, results compared to the control, medical outcomes, study design, study sample size, observations of bias, effect size (Cohen’s *d*), sensitivity, specificity, and F1 (when reported), country of origin, statistics used, patient satisfaction, effectiveness, barriers to adoption, strength of evidence, and quality of evidence.

### 2.7. Study Risk of Bias Assessment

Each reviewer noted observations of bias (e.g., selection bias), and we assessed the quality of each study using the Johns Hopkins Nursing Evidence Based Practice tool (JHNEBP) [14]. These observations were recorded because they affect how to interpret the results, and because bias can limit external validity [15].

### 2.8. Effect Measures

Summary measures were not standardized because we accepted mixed methods and qualitative studies. Measures of effect were summarized in tables for those studies in which it was reported.

### 2.9. Synthesis Methods

Once data extraction was completed, a thematic analysis was performed to make sense of the data. [13] Themes were tabulated and summarized. Results across studies were analyzed for additional inferences and to identify heterogeneity.

### 2.10. Reporting Bias Assessment

We identified the strength and quality of evidence in accordance with the JHNEBP to provide us with an assessment of the applicability of the cumulative evidence and the limit of external validity.

### 2.11. Additional Analyses and Certainty Assessment

We performed a narrative/thematic analysis of the observations to convert them into themes, or common threads between articles. This helped us make sense of the data. We calculated the frequency of occurrence and reported them in affinity matrices. The frequency provided the probability of occurrence in the group of articles analyzed, and it provided confidence in the data analyzed.

### 2.12. Statistical Analysis

Measures of effect were collected during the data extraction process. Where possible, each effect was translated into an effect size equivalent to Cohen’s *d* [16]. These measures were converted into a weighted average effect size by using the sample size for the weight.

## 3. Results

### 3.1. Study Selection

Figure 1 illustrates the study selection process with four databases. A kappa statistic was calculated to estimate the level of agreement between reviewers, (*k* = 0.92, almost perfect agreement) [17,18]. Results from four research databases presented 2021 results. Duplicates and those outside the date range were removed from screening. Using database filters, 1399 records were screened for full text, human subjects, English language, peer-reviewed, and academic journals. Anything except peer-reviewed, published work was excluded along with other systematic literature reviews and meta-analyses. The remaining 68 records were assessed for eligibility. Protocols, editorials, and studies that would not address the objective statement were removed. The remaining group for analysis was 33.

### 3.2. Study Characteristics

PRISMA 2020 and the Kruse Protocol were followed throughout this review. Part of that process is to create a table that lists the characteristics of each study analyzed: participants, intervention, results, medical outcomes, and study design (see Table 1: PICOS). The 33 studies are broken down into the following years: 2012(0), 2013(0), 2014(1) [19], 2015(2) [20,21], 2016(1) [22], 2017(4) [23,24,25,26], 2018(4) [27,28,29,30], 2019(1) [31], 2020(7) [32,33,34,35,36,37,38], 2021(8) [39,40,41,42,43,44,45,46], 2022(5) [47,48,49,50,51]. All studies involved adults as participants. About 76% of the studies were RCT or true experiments, 3 were quasi-experimental, and the rest were a combination of non-experimental, pre-post, qualitative, or mixed methods. About half (16/33, 48%) of the interventions were web-based (eHealth), 13/33 (39%) were mHealth, 3/33 (9%) were telephone-based, and one was a combination of mHealth and eHealth. About 40% of the studies were conducted in the United States, 12% were from Spain, 9% were from the Netherlands, and the rest were from Taiwan, Turkey, Sweden, Norway, India, Iran, and Australia. Almost all studies reported strong positive satisfaction from users, with only one exception [29].

### 3.3. Risk of Bias in and across Studies

Reviewers used the JHNEBP quality assessment tool to identify the strength and quality of evidence. Due to the strong methodologies chosen for review, the JHNEBP tool identified 76% of the articles as Strength I, which means the methodologies were experimental or RCTs (studies had control groups and used randomization). Only 2 studies were identified as Strength II, reserved for quasi-experimental studies. The rest were Strength III, which were a combination of non-experimental, qualitative, observational, pre-post, or mixed methods. Additionally, the JHNEBP tool identified the quality of evidence based on sample size and consistency of evidence. Our group of articles chosen for analysis was 90% (30/33) Quality Q, and only 9% (3/33) were quality B.

### 3.4. Results of Individual Studies

Following the Kruse Protocol, reviewers independently extracted data and recorded observations about each study on a standardized Excel spreadsheet. As part of a thematic analysis, observations that occurred more than once were identified as themes [13]. These themes are tabulated in Table 2. Multiple observations of a similar nature are listed multiple times for studies, but an observation-to-theme match can be found in Appendix A and Appendix B. In 29/33 (88%) studies analyzed, an improvement in at least one area was noted. Additional observations collected in the data extraction step (sample size, bias, effect size, country of origin, statistics used, patient satisfaction, and the strength and quality of evidence from the JHNEBP tool) can be found in Appendix C. Effect sizes were only reported for 22 of the 33 studies (67%). The weighted average effect size was 0.21 (small).

### 3.5. Results of Syntheses, Additional Analysis and Certainty of Evidence

Thematic analysis was performed on all studies. Themes and additional observations were summarized into affinity matrices. Results are sorted by frequency. Frequency is reflected not to imply importance, but only to identify the probability a theme or observation was found in the group of studies analyzed.

#### 3.5.1. Results of Studies Compared with Control Group

Table 3 summarizes the results of the studies compared with a control group. For non-experimental studies, the “no control group” leads the results. This is done to avoid confounding the results. facilitators observed. Thirteen themes and four individual observations were identified by the reviewers for a total of 111 occurrences in the literature. The theme most often observed was “improved mental health”, which occurred 16/111 (14%) occurrences [19,23,34,36,39,40,46,49,50]. This theme combined observations of anxiety, distress, fear of reoccurrence, depression, optimism, self-efficacy, and self-actualization. Sleep outcome was the next most frequently identified theme. It occurred 12/111 (11%) of the occurrences [20,22,28,30,47]. This theme included the following observations: sleep disturbance, insomnia, sleep efficiency, cognitive function, fatigue, and cancer fatigue. The next theme is an improved quality of life, which appeared in 9/111 (8%) of the occurrences [22,27,28,33,35,37,39,44,45]. Two themes appeared in 7/111 (6%) of the occurrences: improved body image [22,31,38,43,45] and improved physical health [27,31,34,44,49]. The body image theme was comprised of the following observations: waist circumference, fat mass, and weight. Two themes were identified in 6/111 (5%) of the occurrences: less numbness, pain, or swelling [22,27,48], and no statistical differences between the intervention and control groups [23,24,26,33,38,41]. Next was less nausea or vomiting [27,34,44]. This occurred in 5/111 (5%) of the observations. Although nausea and vomiting are highly correlated, they are not synonymous, so reviewers chose to report them separately, but they appeared together in two studies. Two themes appeared in 3/111 (3%) of the occurrences: improved global health/return to baseline functioning [22,35,43] and improved social support, and questions were answered by providers [21,29,50]. Two themes occurred in 2/111 (2%) of the occurrences: improved arm symptoms/upper limb functionality [37,48], and the app provided education and answered questions [32,42]. There were four observations that could not be fit into themes: improved exercise, improved medication adherence, improved fasting plasma glucose, and the complexity of the tool (app) takes more time for users to process [25,26,45,46].

#### 3.5.2. Medical Outcome and Effectiveness Commensurate with the Intervention

Table 4 summarizes the medical outcomes and effectiveness observed. Twelve themes and two individual observations were recorded commensurate with the adoption of the intervention for a total of 85 occurrences. Due to the high level of overlap with study results, reviewers chose to only report the differences. In 2/87 (2%) of the occurrences, the intervention was credited with long-term engagement with treatment programs [32,46].

#### 3.5.3. Barriers to the Adoption of Telehealth for Breast Cancer

Table 5 tabulates the barriers identified in the literature. Five themes and one observation were recorded in 49 occurrences. The most frequently observed theme was the need to train users, which occurred in 20/49 (41%) of the occurrences [19,20,27,30,31,32,33,34,35,36,39,40,44,45,46,47,48,49,50,51]. The second barrier was the cost (set up, maintenance, and equipment), which appeared in 18/87 (37%) of the occurrences [22,23,24,25,26,27,29,38,42,43,44,45,46,47,48,49,50,51]. The intervention took time of the providers and presented unusual workflow appeared in 6/49 (12%) of the occurrences [21,28,37,42,43,46]. The intervention was not effective [29,38] or not statistically significant in 2/49 (4%) of the occurrences [40,41]. Finally, there is low reimbursement for the time spent on the intervention that appeared once [21].

#### 3.5.4. Interactions between Observations

The intervention of mHealth resulted in the most observations of “improvement in at least one area”, but not all outcomes were statistically significant [26,27,29,32,34,35,37,39,44,45,47,48,49]. The mHealth intervention studies used strong methodologies: 11 were either RCT or experimental, while one was quasi-experimental and one used mixed methods [26,27,29,32,34,35,37,39,44,45,47,48,49].

## 4. Discussion

This systematic literature review examined 33 studies from 11 countries published over the last 10 years to analyze the effectiveness of telemedicine to treat the symptoms commensurate with the treatment and recovery of breast cancer. Five interventions were identified, however, the dominant interventions were eHealth and mHealth. Methodologies were strong among the group for analysis, and the results of the studies showed positive effects in at least one area [19,21,22,23,24,26,27,28,29,30,31,32,33,34,35,36,37,39,40,42,43,44,45,46,47,48,49,50,51]. Telehealth interventions showed improvements in both mental health [19,23,30,34,36,39,40,46,49,50], physical health [22,24,26,27,31,34,35,43,44,49], sleep outcomes [20,22,28,30,47,51], quality of life [22,27,28,35,37,39,44,45] and body image [24,33,40,45,47]. Telehealth interventions decreased nausea, vomiting [27,34,44], numbness, pain [27,48], improved arm symptoms and upper limb functionality [27,48]. Only a few studies reported non-statistically significant findings [23,24,26,33,38,41].

The findings of this systematic literature review are congruent with that of Buneviciene et al. [10]. The intervention of mHealth and eHealth addressed the quality of life of patients in the areas of physical activity, mindfulness, and stress management. This review found multiple instances of improvements in mental health, physical health, sleep outcomes, and quality of life. Our findings are also consistent with Watanabe et al., in that eHealth and mHealth augmented medical education and health literacy [6].

eHealth and mHealth offer several possible interventions that show promise as a treatment modality of care, however the clinical efficacy of this modality shows mixed results. The difference in results could be due to a difference of methodology or a difference of measurement. While older patients do not often prefer eHealth and mHealth interventions, many other patients do prefer this modality. Even when the results of using the eHealth and mHealth modalities of care show equivalent, but not statistically greater efficacy, offering the modality may meet the preference of the patient. These issues should be addressed in future research considerations.

Future research should examine the reasons for the lack of significant results in some of the studies. Standardization of methodology and measurement should yield consistent results. The results reported in this review were inconsistent. This systematic review focused on breast cancer. Future reviews should examine other types of cancer, then a review of reviews should be conducted for all cancer. The results did not seem to follow any particular intervention. This means it could have been a bias in the sample. Many examples of both sample bias and selection bias were observed, which affect the external and internal validity, respectively.

The results of this review should give practitioners confidence that telehealth can provide viable interventions to help their patients assuage the effects of breast cancer recovery and chemotherapy. The results from the studies analyzed in this review demonstrate healthy habits, less nausea, lost weight, more strength, and an increase in personal confidence. Policy makers should explore other reimbursement mechanisms to ensure the extra time and money these interventions require is reimbursed.

### Limitations

No study is without its limitations, and this literature review is no different. Only four databases were queried over 10 years for published works. A broader scope of databases, years, and sources of literature, such as grey literature, may have identified additional interventions and results. However, the reviewers chose these databases due to their wide availability, 10 years because telemedicine is a rapidly growing field, and published literature to ensure a peer review. Within the studies analyzed were multiple examples of selection and sample bias, which affect the internal and external validity, respectively.

## 5. Conclusions

Telehealth offers promise to help breast cancer survivors cope with the side effects of treatment, the mental anguish that shakes confidence, and the physical ailments that accompany chemotherapy. Several exercise applications show promise educating and helping survivors establish healthy habits to lower the risk of reoccurrence. The most significant barrier is training followed by cost, but these are not significant barriers to overcome.

## Figures and Tables

**Figure 1 healthcare-10-02015-f001:**
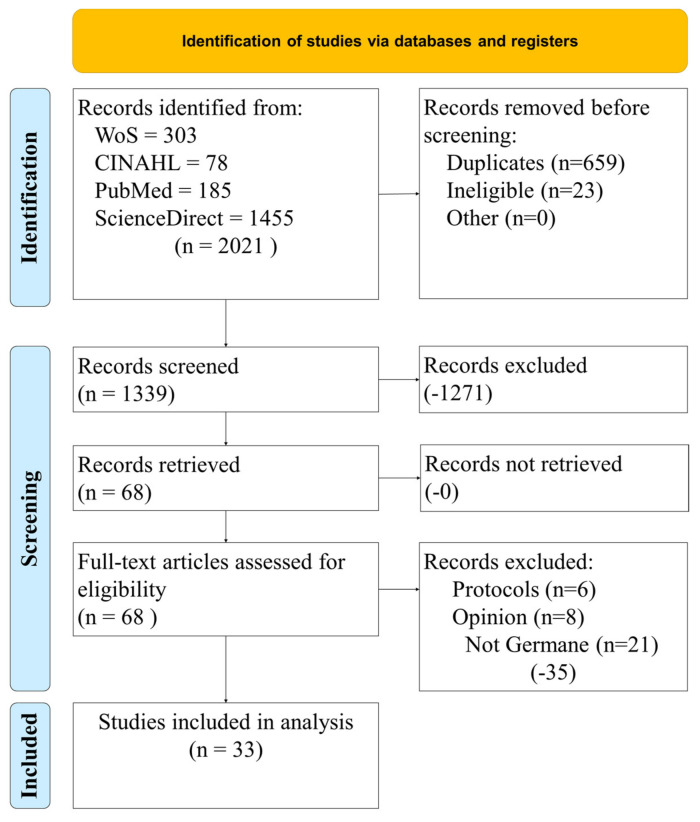
Study selection process.

**Table 1 healthcare-10-02015-t001:** PICOS.

Authors	Participants	Experimental Intervention	Results (Compared to Control Group)	Medical Outcomes Reported	Study Design
Borosund et al. [19]	Adults ≥ 18, avg age 51.4	Internet-based patient-provider communication service	Intervention group reported significantly lower symptom distress, anxiety, and depression	nurse-administered IPPC alone can significantly reduce depression, decreased symptom distress, decreased anxiety	RCT
Freeman et al. [20]	Adults ≥ 18, avg age 55.4	Telemedicine (TD) [vs live vs. wait list]	TD (and Live) reported less fatigue, cognitive dysfunction, and sleep disturbance with WL	improvements in multiple QOL domains for breast cancer survivors compared with WL.Less fatigue, less cognitive dysfunction, fewer sleep disturbances	RCT
Wheelock et al. [21]	Adults ≥ 18, average age 52.85, 73% Caucasian	SIS.NET (online questionnaire with remove NP overview and follow-up)	patients reported more new or changed symptoms compared with standard care patients	This intervention facilitated symptom reporting and may provide a means of convenient symptom assessment	RCT
Galiano-Castillo et al. [22]	Adults ≥ 18	Internet-based, tailored exercise program	telerehabilitation group improved significantly global health status, physical, role, cognitive functioning and arm symptoms, as well as pain severity, and pain interference, compared with the control group.	Improved physical health, cognitive functioning, pain severity, and pain interference	RCT
Admiraal et al. [23]	Adults ≥ 18, average age 53.2	web-based psychoeducation for breast cancer (ENCOURAGE)	No statistically significant differences between control and intervention for optimism or control over future	For clinical distressed patients, use of the intervention increased optimism and control over future	RCT
Fazzino et al. [24]	Adults ≥ 18	telephone (weekly)	No control group. Distance-based weight loss program can be successful	Moderate-to-vigorous physical activity significantly increased from baseline to 6 months.	Non-experimental (no randomization, no control)
Han et al. [25]	Adults ≥ 18, average age 52.2, 88% Caucasian	eHealth system (Comprehensive Health Enhancement Support System, CHESS)	No control group. cancer patients’ access to more complex tools generates more use with their time spreading out over the diverse services.	Communication functions drive long-term engagement with the system.	Pre-post
Uhm et al. [26]	Adults ≥ 18	mHealth	Improved exercise, but not statistically different than control	Improved exercise, but not statistically different than control	Quasi-experimental
Kim et al. [27]	Adults ≥ 18	mHealth (mobile game)	Improved drug adherence, lower side effects of chemotherapy (nausea, fatigue, numbness of hand or foot, and hair loss). Improved quality of life. No significant difference in depression or anxiety	Improved drug adherence, lower side effects of chemotherapy (nausea, fatigue, numbness of hand or foot, and hair loss). Improved quality of life. Improved medication adherence. No significant difference in depression or anxiety	RCT
McCarthy et al. [28]	Adults ≥ 18	nurse-led telemedicine delivered, cognitive behavioral therapy	participants reported improvements in sleep outcomes, including SE and SL. QOL and daily functioning improved, but anxiety and depression did not.	participants reported improvements in sleep outcomes, including SE and SL. QOL and daily functioning improved, but anxiety and depression did not.	Quasi-experimental
Visser et al. [29]	Adults ≥ 18	tablet online support group	No statistically significant differences between control and intervention for distress and empowerment. Greater peer support identified in control.	No improvement with intervention. Satisfaction very low.	RCT
Zachariae et al. [30]	Adults ≥ 18, average age 52.3	Internet-delivered cognitive-behavioral therapy (iCBT)	Statistically significant improvements observed for all sleep-related outcomes (fatigue, sleep disturbances, total sleep time).	Reduced insomnia, increased sleep quality, increases sleep efficiency, increased total sleep time, improved time in bed, reduced fatigue	RCT
Ariza-Garcia et al. [31]	Adults ≥ 18	web-based exercise system (e_CuidateChemo)	Functional capacity improved significantly, abdominal strength, lower body strength, back strength	Intervention increased exercise capacity by 10.8% (33.4% reached a normal exercise capacity compared with 12.3% in control). Functional capacity, abdominal strength, lower body strength, back strength improved significantly.	RCT
Crafoord et al. [32]	Adults ≥ 18	mHealth app for symptom self-management	Daily symptom reporting created feelings of having continuous contact with health care professionals, being acknowledged, and safe.	Engagement was very high for intervention. The app promoted patient participation in their care.	Mixed Methods
Ferrante et al. [33]	Adults ≥ 60, African American only	mHealth/eHealth tools	No statistically significant differences between weight lost in both groups. Waist circumference improved more, quality of life more, and use of strategies for healthy eating and decreasing calories.	Effective at weight loss, but not statistically significant	RCT
Fjell et al. [34]	Adults ≥ 18, average age 48	mHealth app (Interaktor) during neoadjuvant chemo	statistically significant less symptom prevalence in nausea, vomiting, feeling sad, appetite loss and constipation. Overall symptom distress and physical symptom distress were rated statistically significant lower in the intervention group. Further, emotional functioning was rated statistically significant higher in the intervention group.	statistically significant less symptom prevalence in nausea, vomiting, feeling sad, appetite loss and constipation. Overall symptom distress and physical symptom distress were rated statistically significant lower in the intervention group. Further, emotional functioning was rated statistically significant higher in the intervention group.	RCT
Hou et al. [35]	Adults ≥ 50	mHealth app for self-management support (BCSMS)	Mean quality of life scores and global health higher	Mean quality of life scores and global health higher	RCT
Lally et al. [36]	Adults ≥ 18	we-based, psychoeducational distress self-management program (CaringGuidance)	post hoc analysis showed significant group differences in slopes occurring between study months 2 and 3 on distress and depressive symptoms	post hoc analysis showed significant group differences in slopes occurring between study months 2 and 3 on distress and depressive symptoms	True experiment
Lozano-Lozano et al. [37]	Adults ≥ 18	mHealth (BENECA) + rehab	Both groups showed improved outcomes, but global QoL was significantly better with intervention. Improvement in upper-limb functionality also higher	Both groups showed improved outcomes, but global QoL was significantly better with intervention. Improvement in upper-limb functionality also higher	RCT
van der Hout et al. [38]	Adults ≥ 56	eHealth (Oncokompas) symptom self-management app	Oncokompas did not improve the amount of knowledge, skills, and confidence for self-management in cancer survivors.	No difference between groups	RCT
Çınar et al. [39]	Adults ≥ 18	mHealth app for education, symptom tracking, and management	QoL of the treatment group after intervention increased and distress level was lower	QoL of the treatment group after intervention increased and distress level was lower	True experiment
Fang et al. [40]	Adults ≥ 20	decision-support app (Pink Journey)	body image distress declined significantly for the intervention group but increased for the control group. no significant difference in decision conflict, decision regret, anxiety, or depression.	Decrease in body image, regret, anxiety, & distress	RCT
Krzyzanowska et al. [41]	Adults ≥ 40	telephone based management of toxicities	No differences in self-efficacy, anxiety, or depression	No differences in self-efficacy, anxiety, or depression	RCT
Kumar et al. [42]	Adult, aged 27	Teleconsultation	No control group. Concerns and questions answered through intervention	Breast conservation surgery	Qualitative
Lai et al. [43]	Adults ≥ 18, avg age 56.8, 53% Caucasian	Telemedicine (VTC) Occupational Therapy	No control group. Patients regained baseline function within a mean of 42.4 days after surgery and after an average of three sessions	all regained baseline functional status and full range of motion	Non-experimental (no randomization, no control)
Öztürk et al. [44]	Adults ≥ 18	mHealth symptom monitoring app	Effective at decreasing nausea-vomiting, raising sexual function and sexual enjoyment	Symptom monitoring with mHealth highly effective in controlling physical symptoms	True experiment
Reeves et al. [45]	Adults ≥ 45	mHealth weight-loss	Improved weight reduction (over control) fat mass, metabolic syndrome risk score, waist circumference, fasting plasma glucose, and quality of life	Improved weight reduction (over control) fat mass, metabolic syndrome risk score, waist circumference, fasting plasma glucose, and quality of life	RCT
Wagner et al. [46]	Adults ≥ 18	eHealth (Fear of recurrence, FoR) Telecoaching	Significantly reduced fear of recurrence. Telecoaching improved adherence and retention.	Reduced fear of recurrence. Telecoaching improved adherence and retention.	RCT
Bandani-Susan et al. [47]	Adults ≥ 18, average age 46.34	mHealth education	Mean score of cancer fatigue decreased and body image increased significantly	Decreased fatigue, increased body image	RCT
Fu et al. [48]	Adults ≥ 18	mHealth pain-management	Participants in the intervention were more likely to experience complete reduction in pain and soreness, lower median severity scores and general body pain, less arm/hand swelling, heaviness, redness, and limited movement in shoulder	Less pain, less soreness, less swelling, less heaviness, less redness, less limited movement in shoulder	RCT
Gao et al. [49]	Adults ≥ 18, average age 56.17	mHealth Tai Chi and health education	A significant time effect for mental health, physical health, but not for stress.	Tai Chi participants had a significantly better mental health at follow up.	RCT
Medina et al. [50]	Adults ≥ 18, average age 52.35	eHealth ecosystem (ICOnnecta)	Strong social support led to better psychosocial course	ICOnnecta supports the development of a digital relation with healthcare services	Quasi-experimental
Oswald et al. [51]	Adults ≥ 18	eHealth cognitive-behavioral therapy (iCBT)	Improvements in insomnia, sleep efficiency, and sleep disturbance	Improvements in insomnia, sleep efficiency, and sleep disturbance	RCT

BCMSM: Breast cancer self-management support; CHESS: Comprehensive Health Enhancement Support System; FoR: Fear of reoccurrence; QoL: Quality of Life; iCBT: Internet Cognitive Behavior Therapy; IPPC: Internet-based provider communications service; SIS:NET: System for Individualized Survivorship Care; SE: Sleep efficiency; SL: Sleep latency; TD: Telemedicine delivery; VTC: Video tele-conference; WL: Wait list.

**Table 2 healthcare-10-02015-t002:** Summary of analysis, sorted chronologically.

Authors	Intervention Themes	Results Themes	Medical Outcome Themes	Effectiveness Themes	Barrier Themes
Borosund et al. [19]	Web-based (eHealth)	Improved in at least one area	Improved mental health	Improved mental health	Must train users
		Improved mental health	Improved mental health	Improved mental health	
		Improved mental health			
Freeman et al. [20]	Web-based (eHealth)	Improved sleep outcomes	Improved sleep outcomes	Improved sleep outcomes	Must train users
Wheelock et al. [21]	Web-based (eHealth)	Improved in at least one area	Provided education/answered questions	Provided education/answered questions	Time of providers/workflow
		Low reimbursement of treatment			Improved social support/answered questions
Galiano-Castillo et al. [22].	Web-based (eHealth)	Improved in at least one area	Improved physical health	Improved physical health	Cost of intervention
		Improved global health/baseline function	Improved sleep outcomes	Improved sleep outcomes	
		Improved sleep outcomes	Less pain	Less pain	
		Less numbness/pain/swelling	Improved quality of life	Improved quality of life	
		Improved quality of life			
Admiraal et al. [23]	Web-based (eHealth)	Improved in at least one area	Improved mental health	Improved mental health	Cost of intervention
		Improved mental health	Improvements not statistically significant	Improvements not statistically significant	
		No statistically significant differences			
Fazzino et al. [24]	Telephone	Improved in at least one area	Improved physical health	Improved physical health	Cost of intervention
		Improved body image	Improved body image	Improved body image	
		No statistically significant differences	Improvements not statistically significant	Improvements not statistically significant	
Han et al. [25]	Web-based (eHealth)	Complexity of tool takes more time to process	Provided education/answered questions	Provided education/answered questions	Cost of intervention
Uhm et al. [26]	mHealth	Improved in at least one area	Improved physical health	Improved physical health	Cost of intervention
		Improved exercise	Improvements not statistically significant	Improvements not statistically significant	
		No statistically significant differences			
Kim et al. [27]	mHealth	Improved in at least one area	Less nausea/vomiting	Improved medication adherence	Cost of intervention
		Less nausea/vomiting	Less numbness	Less nausea/vomiting	Must train users
		Less numbness/pain/swelling	Improved physical health	Improved sleep outcomes	
		Improved physical health	Improved quality of life	Less numbness	
		Improved quality of life	Improved medication adherence	Improved quality of life	
McCarthy et al. [28]	Web-based (eHealth)	Improved in at least one area	Improved sleep outcomes	Improved sleep outcomes	Time of providers/workflow
		Improved sleep outcomes	Improved quality of life	Improved quality of life	
		Improved quality of life			
Visser et al. [29]	mHealth	Improved in at least one area	Provided education/answered questions	Improvements not statistically significant	Intervention not effective
		Improved social support/answered questions			Cost of intervention
Zachariae et al. [30]	Web-based (eHealth)	Improved in at least one area	Improved sleep outcomes	Improved sleep outcomes	Must train users
		Improved sleep outcomes	Improved sleep outcomes	Improved sleep outcomes	
		Improved sleep outcomes	Improved sleep outcomes	Improved sleep outcomes	
		Improved sleep outcomes	Improved mental health	Improved mental health	
Ariza-Garcia et al. [31]	Web-based (eHealth)	Improved in at least one area	Improved physical health	Improved physical health	Must train users
		Improved physical health	Improved physical health	Improved physical health	
		Improved physical health	Improved physical health	Improved physical health	
		Improved physical health	Improved physical health	Improved physical health	
Crafoord et al. [32]	mHealth	Improved in at least one area	long-term engagement with intervention	long-term engagement with intervention	Must train users
		Provided education/answered questions	Provided education/answered questions	Provided education/answered questions	
Ferrante et al. [33]	mHealth + eHealth	Improved in at least one area	Improved body image	Improved physical health	Must train users
		Improved body image	Improvements not statistically significant	Improved body image	
		Improved quality of life		Improved quality of life	
		No statistically significant differences			
Fjell et al. [34]	mHealth	Improved in at least one area	Less nausea/vomiting	Less nausea/vomiting	Must train users
		Less nausea/vomiting			
		Less nausea/vomiting	Less nausea/vomiting	Less nausea/vomiting	
		Improved mental health	Improved mental health	Improved mental health	
		Improved mental health	Improved mental health	Improved mental health	
		Improved physical health	Improved physical health	Improved physical health	
Hou et al. [35]	mHealth	Improved in at least one area	Improved quality of life	Improved quality of life	Must train users
		Improved quality of life	Improved physical health	Improved physical health	
		Improved global health/baseline function			
Lally et al. [36]	Web-based (eHealth)	Improved in at least one area	Improved mental health	Improved mental health	Must train users
		Improved mental health	Improved mental health	Improved mental health	
		Improved mental health			
Lozano-Lozano et al. [37]	mHealth	Improved in at least one area	Improved quality of life	Improved quality of life	Time of providers/workflow
		Improved quality of life	Improved arm symptoms/upper limb functionality	Improved arm symptoms/upper limb functionality	
		Improved arm symptoms/upper limb functionality			
van der Hout et al. [38]	Web-based (eHealth)	No statistically significant differences	Improvements not statistically significant	Improvements not statistically significant	Intervention not effective
					Cost of intervention
Çınar et al. [39]	mHealth	Improved in at least one area	Improved quality of life	Improved quality of life	Must train users
		Improved quality of life	Improved mental health	Improved mental health	
		Improved mental health			
Fang et al. [40]	Web-based (eHealth)	Improved in at least one area	Improved body image	Improved body image	Intervention not statistically effective
		Improved body image	Improved mental health	Improved mental health	Must train users
		Improved mental health	Improved mental health	Improved mental health	
		Improved mental health	Improved mental health	Improved mental health	
		Improved mental health			
Krzyzanowska et al. [41]	Telephone	No statistically significant differences	Improvements not statistically significant	Improvements not statistically significant	Intervention not statistically effective
Kumar et al. [42]	Telephone	Improved in at least one area	Provided education/answered questions	Provided education/answered questions	Cost of intervention
		Provided education/answered questions			Time of providers/workflow
Lai et al. [43]	Web-based (eHealth)	Improved in at least one area	Improved physical health	Provided education/answered questions	Cost of intervention
		Improved global health/baseline function			Time of providers/workflow
Öztürk et al. [44]	mHealth	Improved in at least one area	Less nausea/vomiting	Less nausea/vomiting	Cost of intervention
		Less nausea/vomiting	Less nausea/vomiting	Less nausea/vomiting	Must train users
		Less nausea/vomiting	Improved quality of life	Improved quality of life	
		Improved quality of life	Improved physical health	Improved physical health	
		Improved physical health			
Reeves et al. [45]	mHealth	Improved in at least one area	Improved body image	Improved body image	Cost of intervention
		Improved body image	Improved body image	Improved body image	Must train users
		Improved body image	Improved body image	Improved body image	
		Improved body image	Improved fasting plasma glucose	Improved fasting plasma glucose	
		Improved fasting plasma glucose	Improved quality of life	Improved quality of life	
		Improved quality of life			
Wagner et al. [46]	Web-based (eHealth)	Improved in at least one area	Improved mental health	Improved mental health	Cost of intervention
		Improved mental health	long-term engagement with intervention	long-term engagement with intervention	Time of providers/workflow
		Improved medication adherence			Must train users
Bandani-Susan et al. [47]	mHealth	Improved in at least one area	Improved sleep outcomes	Improved sleep outcomes	Cost of intervention
		Improved sleep outcomes	Improved body image	Improved body image	Must train users
		Improved body image			
Fu et al. [48]	mHealth	Improved in at least one area	Less pain	Less pain	Cost of intervention
		Less numbness/pain/swelling	Less pain	Less pain	Must train users
		Less numbness/pain/swelling	Less pain	Less pain	
		Less numbness/pain/swelling	Less numbness	Less numbness	
		Less numbness/pain/swelling	Improved arm symptoms/upper limb functionality	Improved arm symptoms/upper limb functionality	
		Improved arm symptoms/upper limb functionality			
Gao et al. [49]	mHealth	Improved in at least one area	Improved mental health	Improved mental health	Cost of intervention
		Improved mental health	Improved physical health	Improved physical health	Must train users
		Improved physical health			
Medina et al. [50]	Web-based (eHealth)	Improved in at least one area	Improved mental health	Improved mental health	Cost of intervention
		Improved social support/answered questions			Must train users
		Improved mental health			
Oswald et al. [51]	Web-based (eHealth)	Improved in at least one area	Improved sleep outcomes	Improved sleep outcomes	Cost of intervention
		Improved sleep outcomes	Improved sleep outcomes	Improved sleep outcomes	Must train users
		Improved sleep outcomes	Improved sleep outcomes	Improved sleep outcomes	
		Improved sleep outcomes			

**Table 3 healthcare-10-02015-t003:** Results of studies, compared to control group.

Results Themes and Observations	Frequency
Improved in at least one area [19,21,22,23,24,26,27,28,29,30,31,32,33,34,35,36,37,39,40,42,43,44,45,46,47,48,49,50,51]	29
Improved mental health [19,23,34,36,39,40,46,49,50]	16
Improved sleep outcomes [20,22,28,30,47]	12
Improved quality of life [22,27,28,33,35,37,39,44,45]	9
Improved body image [24,33,40,45,47]	7
Improved physical health [27,31,34,44,49]	7
Less numbness/pain/swelling [22,27,48]	6
No statistically significant differences [23,24,26,33,38,41]	6
Less nausea/vomiting [27,34,44]	5
Improved global health/baseline function [22,35,43]	3
Improved social support/answered questions [21,29,50]	3
Improved arm symptoms/upper limb functionality [37,48]	2
Provided education/answered questions [32,42]	2
Improved exercise [26]	1
Improved medication adherence [46]	1
Improved fasting plasma glucose [45]	1
Complexity of tool takes more time to process [25]	1
	111

**Table 4 healthcare-10-02015-t004:** Medical outcomes and effectiveness commensurate with the adoption of the intervention.

Medical Outcomes and Effectiveness Themes and Observations	Frequency
Improved mental health [19,23,30,34,36,39,40,46,49,50]	17
Improved physical health [22,24,26,27,31,34,35,43,44,49]	13
Improved sleep outcomes [20,22,28,30,47,51]	12
Improved quality of life [22,27,28,35,37,39,44,45]	8
Improved body image [24,33,40,45,47]	7
Improvements not statistically significant [23,24,26,33,38,41]	6
Less nausea/vomiting [27,34,44]	5
Provided education/answered questions [21,25,29,32,42]	5
Less pain [22,48]	4
Less numbness [27,48]	2
Improved arm symptoms/upper limb functionality [37,48]	2
long-term engagement with intervention [32,46]	2
Improved medication adherence [27]	1
Improved fasting plasma glucose [45]	1
	85

**Table 5 healthcare-10-02015-t005:** Barriers to the adoption of Telehealth for the treatment of Breast Cancer.

Barrier Themes and Observations	Frequency
Must train users [19,20,27,30,31,32,33,34,35,36,39,40,44,45,46,47,48,49,50,51]	20
Cost of intervention [22,23,24,25,26,27,29,38,42,43,44,45,46,47,48,49,50,51]	18
Time of providers/workflow [21,28,37,42,43,46]	6
Intervention not effective [29,38]	2
Intervention not statistically effective [40,41]	2
Low reimbursement of treatment [21]	1
	49

## Data Availability

Data from this study can be obtained by asking the lead author.

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
