# Peer review of "Leveraging Telehealth for the Management of Breast Cancer: A Systematic Review"

_healthcare, 2022, doi:10.3390/healthcare10102015_

Round 1

Reviewer 1 Report

First of all, I want to note that it has been a pleasure review your manuscript. I think this is an interesting topic for clinicians who manage this prevalent condition.

This study  provides an insight into the efficacy of Telehealth for breast cancer.

In order to improve the quality of the manuscript. After reading in depth the manuscript, I would like to make some comments and ask the authors several questions about.

- In the section on affiliations, there is a mistake in the formatting

- A paragraph would be missing in the Introduction section to emphasise why this review is necessary.

- Page 2, lines 68-70: “Studies were collected and analyzed from the 68 last 10 years if they were published in the English language in peer-reviewed, academic 69 journals and included adults as participants”.  This paragraph does not belong in the introduction section, but rather in the Methods section.

- Page 2.  Line 87: MEDLINE was excluded from all databases except PubMed. It is not understood why MEDLINE was excluded.

- Correctly spell the word at the end of the line 94, page 3.

- Figure 1 states: records excluded: Not German. This is not in line with the eligibility criteria: “Only English is mentioned: To be included in the group of articles for analysis, studies had to be published in the English language in the last 10 years in peer-reviewed, academic journals, and used human adult females as subjects. To avoid confounding results, other reviews were excluded”.

- Page 4, line 142: PRISMA in capital letters.

- table 1 is not in the journal format. In row 1 there is a dash that should not be there.

- Below table 1, the legend explaining each acronym used in the table is missing.

- Tabla 1: Öztürk et al. [42]: The type of study is not understood: True experiment.

- Tabla 1: Lai et al. [41]: Correctly terminate the design type column.

- “Table 2: Summary of analysis, sorted chronologically”ç. Identify table 2, just before the table, not after.

In row 1, mental health is repeated many times.

- Oswald et al. [49]. It is also repeated “sleep outcomes”.

 If you would be so kind, please review the possible mistakes in the table and adapt it to the format of the journal. This should be done for all tables in the manuscript.

- Please review the entire text so that the words end correctly: e.g.: lines 276, 277..

- The discussion is short. The first part is limited to stating the results. The discussion section should be improved. A more direct discussion of the general findings is needed.

Author Response

Reviewer 1

First of all, I want to note that it has been a pleasure review your manuscript. I think this is an interesting topic for clinicians who manage this prevalent condition.

This study  provides an insight into the efficacy of Telehealth for breast cancer.

In order to improve the quality of the manuscript. After reading in depth the manuscript, I would like to make some comments and ask the authors several questions about.

- In the section on affiliations, there is a mistake in the formatting

**On Track-Changes, added the Street, City, Zip

- A paragraph would be missing in the Introduction section to emphasise why this review is necessary.

** I added more information in the paragraph from lines 49-57 to justify the study.

- Page 2, lines 68-70: “Studies were collected and analyzed from the 68 last 10 years if they were published in the English language in peer-reviewed, academic 69 journals and included adults as participants”.  This paragraph does not belong in the introduction section, but rather in the Methods section.

**Eliminated this sentence, since it is repeated again in the Methods section below

- Page 2.  Line 87: MEDLINE was excluded from all databases except PubMed. It is not understood why MEDLINE was excluded.

**PubMed includes citations indexed under MEDLINE. We eliminated MEDLINE from the other databases in order to eliminate duplicates.

- Correctly spell the word at the end of the line 94, page 3.

**Analysis was the word in question. It was hyphenated before but the document (now on line 93) now shows “analysis” in one single line.

- Figure 1 states: records excluded: Not German. This is not in line with the eligibility criteria: “Only English is mentioned: To be included in the group of articles for analysis, studies had to be published in the English language in the last 10 years in peer-reviewed, academic journals, and used human adult females as subjects. To avoid confounding results, other reviews were excluded”.

**I believe the word to which you refer is "Not germane" which means (not applicable), as opposed to German (a language).

- Page 4, line 142: PRISMA in capital letters.

**Thank you, corrected (tracked change on Line 144)

- table 1 is not in the journal format. In row 1 there is a dash that should not be there.

Could not find.... Saw (after line 161) “Internet-based” “patient-provider”... 

** Formatted table to delete the column

- Below table 1, the legend explaining each acronym used in the table is missing.

** Added the acronyms in the last line of the table

- Tabla 1: Öztürk et al. [42]: The type of study is not understood: True experiment.

** This refers to a true experiment as opposed to quasi-experimental. The former pertains to a study design in which there is random assignment into either the control or the exposure group and equal measures on the two groups for comparison.

- Tabla 1: Lai et al. [41]: Correctly terminate the design type column.

**This study did not provide a randomization and there was no control group, thereby classifying as non-experimental study design.

- “Table 2: Summary of analysis, sorted chronologically”ç. Identify table 2, just before the table, not after.

**Updated: labeled table in line 180 (track changes).

In row 1, mental health is repeated many times.

**This was explained in the paragraph above Table 2: “Multiple observations of a similar nature are listed multiple times for studies. . .” (lines 174-175). We provided an appendix for an observation-to-theme match.

- Oswald et al. [49]. It is also repeated “sleep outcomes”.

**This was explained in the paragraph above Table 2: “Multiple observations of a similar nature are listed multiple times for studies. . .” (lines 174-175). We provided an appendix for an observation-to-theme match.

 If you would be so kind, please review the possible mistakes in the table and adapt it to the format of the journal. This should be done for all tables in the manuscript.

** We reviewed the tables and corrected errors.

- Please review the entire text so that the words end correctly: e.g.: lines 276, 277..

No hyphenation (now line 294).

**I believe you are referring to the hyphenation of words when they span multiple lines. This was not done by the authors: It was done by the type-setting team of the journal when they took our manuscript and inserted it into the journal’s format.

- The discussion is short. The first part is limited to stating the results. The discussion section should be improved. A more direct discussion of the general findings is needed.

** We improved the discussion section. Please note, a summary of the findings is required under the PRISMA 2020 checklist.

Reviewer 2 Report

Comments:

Overall, the approach herein seems distinguished and provides adequate background for the manuscript design based on previous studies; the information sources and the content analysis are also suitable for understanding telemedicine concepts. A statistical data analysis section would have made this an excellent MS – and clarified how authors could plan to use the published information from previous studies to write a review article.

The authors made an excellent reference to the theoretical fundament for MS. However, the rationale for “telemedicine” seems superficial and needs a more logical explanation. More discussion about the limitations of telemedicine from the perspective of eHealth and mHealth would have strengthened the MS. The information taken from the four databases is derived and sustained by the literature review and is relevant to the proposed idea. However, authors may incorporate information from more databases.

Specific comments are as follows.

1.     Why did the authors choose only breast cancer patients as telehealth is a general term that applies to all diseases? Was there any research gap between breast cancer and telehealth? Can’t telemedicine be appropriate for all cancer patients as it works using information and communications technologies? mHealth and eHealth stand for what? It should be defined in the MS.

2.     The author has mentioned that to use telemedicine, the information is collected from different sources like mobile apps, etc. how authors determine the authenticity of the information taken from mobile apps, text messages through SMS, telephonic calls, websites, and computer programs. What are the criteria for data validation?

3.     The method used is generally appropriate; however, clarification of a few details and providing a rationale for this method should be incorporated clearly. 

The PRISMA 2020 guidelines for systematic review.

The method part is not followed according to the guidelines 

For example:

Eligibility criteria: The clear characteristic requirement for the study used to decide the eligibility for inclusion is unclear. Specific reasons for the exclusion of the studies were not clear. “To avoid confounding results” is very vague.

Information sources:

Were all the articles freely available in full text? Contacted any authors/individuals for data?

Search strategy:

Are any automated tools involved?

Any validated method was used?

Selection process:

How many reviewers screened the studies? How many are included, and how many are excluded by each reviewer?

All the sections are a bit vague and not following the guidelines. 

General comments:

1.     Abstract L12 “globally around the world” Globally is the world only. It should be globally only. 

2.     Abstract L13 “Telemedicine has been used to treat the symptoms.” Telemedicine is not a drug, so how can it be used to treat the symptoms associated with BC? It can help clinicians to prescribe the medication and ultimately help treat BC patients. 

3.     Abstract L14 “To analyze the effectiveness of telemedicine to help women recover from the treatment.” It’s a very wrong statement or wrongly written. How can you recover a patient from the treatment? Treatment is given to the patient to get recover from the condition she is having. You can retrieve the patients from the side effects of the treatment.  

4.     Abstract L18 “Five interventions were identified in the literature, with the most dominant being eHealth and mHealth.” What are the other three interventions? 

5.     Discussion L240 “Four interventions were identified.” Previously in the abstract, the authors mentioned that they identified five interventions, whereas in the discussion part, the authors writing they had identified four interventions. Doesn’t match the statements?  

6.     Discussion L256 “The studies analyzed in this review demonstrate healthy habits, less nausea, lost weight, more strength, and personal confidence.” It’s a very vague written sentence, and the primary message or context is not clear.

7.     Authors mention, “One form of telemedicine is mHealth and eHealth.” Having said one of the forms, how do they mention two terms in the sentence?

8.     SMS is a “short message service,” wrongly written in the manuscript as a “simple message system.”

9.     L87: MEDLINE was excluded from all databases except PubMed. What does the author mean by this statement?

10. Proofreading is strongly recommended.

Author Response

Reviewer 2

Comments:

Overall, the approach herein seems distinguished and provides adequate background for the manuscript design based on previous studies; the information sources and the content analysis are also suitable for understanding telemedicine concepts. A statistical data analysis section would have made this an excellent MS – and clarified how authors could plan to use the published information from previous studies to write a review article.

** We added a statistical analysis section. We also calculated and reported the weighted average effect size.

The authors made an excellent reference to the theoretical fundament for MS. However, the rationale for “telemedicine” seems superficial and needs a more logical explanation. More discussion about the limitations of telemedicine from the perspective of eHealth and mHealth would have strengthened the MS. The information taken from the four databases is derived and sustained by the literature review and is relevant to the proposed idea. However, authors may incorporate information from more databases.

Specific comments are as follows.

  1. Why did the authors choose only breast cancer patients as telehealth is a general term that applies to all diseases? Was there any research gap between breast cancer and telehealth? Can’t telemedicine be appropriate for all cancer patients as it works using information and communications technologies? mHealth and eHealth stand for what? It should be defined in the MS.

** Thank you for the comments. I will attempt to address all of your comments.

  1. While telehealth is a modality of care for all diseases, it is impossible to study all diseases. We must take conditions, diseases, environments of care, etc. piecemeal in order to properly analyze each one. Once a large compendium of studies exist across spectrums of care and diseases, a review of reviews can be conducted.
  2. There is a research gap between breast cancer and telehealth. We added a couple sentences in the introduction to highlight this gap.
  3. Yes, telemedicine is appropriate as a modality for all cancer care, but all cancer care is too large to study. We actually started with that scope, but the results were in the hundreds of thousands. We had to narrow the scope. We plan to examine several aspects of cancer followed by a review of reviews.
  4. eHealth and mHealth were defined in the introduction on line 45.
  5. The author has mentioned that to use telemedicine, the information is collected from different sources like mobile apps, etc. how authors determine the authenticity of the information taken from mobile apps, text messages through SMS, telephonic calls, websites, and computer programs. What are the criteria for data validation?

** The JHNEBP tool does not analyze veracity of data. We used this quality measurement tool to analyze what was reported (consistency of results, appropriate size of samples, adequate controls, etc.). We restricted the search to research published in peer-reviewed journals. The source of publication is responsible for verifying veracity of data.

  1. The method used is generally appropriate; however, clarification of a few details and providing a rationale for this method should be incorporated clearly.

The PRISMA 2020 guidelines for systematic review.

** We provided some explanation of this standard.

The method part is not followed according to the guidelines

For example:

Eligibility criteria: The clear characteristic requirement for the study used to decide the eligibility for inclusion is unclear. Specific reasons for the exclusion of the studies were not clear. “To avoid confounding results” is very vague.

** We provided explanation for this practice.

Information sources:

Were all the articles freely available in full text? Contacted any authors/individuals for data?

** We were able to find all articles either in our library or through interlibrary loan. We did not need to contact authors for their data.

Search strategy:

Are any automated tools involved?

** We used only the search tools available at each database.

Any validated method was used?

** We reported all methodology collected. We did not allow editorials or expert opinions. These would not be studies. We stated, “. . . studies had to be published . . . “ implying that we did not accept grey literature, editorials, opinions, or unpublished works.

Selection process:

How many reviewers screened the studies? How many are included, and how many are excluded by each reviewer?

** Abstracts were screened by at least two reviewers. This is now stated on line 100.

All the sections are a bit vague and not following the guidelines.

** We added clarification into most sections. If the reviewer can be more specific, we can address all concerns.

General comments:

  1. Abstract L12 “globally around the world” Globally is the world only. It should be globally only.

** “around the world” deleted.

  1. Abstract L13 “Telemedicine has been used to treat the symptoms.” Telemedicine is not a drug, so how can it be used to treat the symptoms associated with BC? It can help clinicians to prescribe the medication and ultimately help treat BC patients.

** Great point. We clarified this is a modality of care, not the care itself.

  1. Abstract L14 “To analyze the effectiveness of telemedicine to help women recover from the treatment.” It’s a very wrong statement or wrongly written. How can you recover a patient from the treatment? Treatment is given to the patient to get recover from the condition she is having. You can retrieve the patients from the side effects of the treatment.

**I disagree. The treatment of cancer is the very definition of double-effect. We introduce poison and radiation into the body in the hopes that the treatment will be less than death. This treatment requires recovery. Telemedicine is used to help cancer survivors cope with the treatment as well as the disease.

  1. Abstract L18 “Five interventions were identified in the literature, with the most dominant being eHealth and mHealth.” What are the other three interventions?

** Good point. We added the others.

  1. Discussion L240 “Four interventions were identified.” Previously in the abstract, the authors mentioned that they identified five interventions, whereas in the discussion part, the authors writing they had identified four interventions. Doesn’t match the statements?

** Good point. We corrected L240 to be five interventions.

  1. Discussion L256 “The studies analyzed in this review demonstrate healthy habits, less nausea, lost weight, more strength, and personal confidence.” It’s a very vague written sentence, and the primary message or context is not clear.

** The PRISMA 2020 standard requires a summary of the findings. This sentence summarizes the themes identified. We added “an increase in” for personal confidence. That way it fits the parallelism of the sentence.

  1. Authors mention, “One form of telemedicine is mHealth and eHealth.” Having said one of the forms, how do they mention two terms in the sentence?

** This is addressed in the introduction. Because so many apps can be accessed on mobile devices, the lines between mHealth and eHealth have become quite blurred. Researchers might use an eHealth app, but they do not track whether or not it was accessed through a computer or through a mobile device. Distinguishing the two has become difficult.

  1. SMS is a “short message service,” wrongly written in the manuscript as a “simple message system.”

** Great point. We corrected this error.

  1. L87: MEDLINE was excluded from all databases except PubMed. What does the author mean by this statement?

** We added a couple sentences for clarification. This means we excluded MEDLINE from all databases except PubMed in order to eliminate duplicates. PubMed provided us with all MEDLINE studies.

  1. Proofreading is strongly recommended.

** We proofed the manuscript once mo

Round 2

Reviewer 2 Report

Only one comment; authors should revise the following statement to;

Line 14: "To analyze..........recover from the treatment-associated effects and promote overall recovery from breast cancer."

Reason: The treatment is an independent entity, its action is not dependent on telemedicine or any other approach. However, telemedicine-based approaches provide support to patients.

Author Response

Line 14: "To analyze..........recover from the treatment-associated effects and promote overall recovery from breast cancer."

Reason: The treatment is an independent entity, its action is not dependent on telemedicine or any other approach. However, telemedicine-based approaches provide support to patients.

**We concur with the suggested change, and we have made this change. Thank you for the input. Your feedback makes our work stronger.

Dr. Kruse